# Non-Invasive Biomarkers for Early Diagnosis of Kidney Allograft Dysfunction: Current and Future Applications in the Era of Precision Medicine

**DOI:** 10.3390/medicina61020262

**Published:** 2025-02-04

**Authors:** Christina Lazarou, Eleni Moysidou, Michalis Christodoulou, Georgios Lioulios, Erasmia Sampani, Chrysostomos Dimitriadis, Asimina Fylaktou, Maria Stangou

**Affiliations:** 1Department of Nephrology, Papageorgiou General Hospital, 56429 Thessaloniki, Greece; lachristine91@yahoo.com; 2School of Medicine, Aristotle University of Thessaloniki, 54642 Thessaloniki, Greece; moysidoueleni@yahoo.com (E.M.); michalischristodoulou22@gmail.com (M.C.); sampaniedoc@gmail.com (E.S.); chrydim@gmail.com (C.D.); 3Department of Nephrology, 424 Military Hospital of Thessaloniki, 56429 Thessaloniki, Greece; pter43@yahoo.gr; 4Department of Immunology, National Peripheral Histocompatibility Center, General Hospital Hippokration, 54642 Thessaloniki, Greece; fylaktoumina@gmail.com; 51st Department of Nephrology, Hippokration Hospital, School of Medicine, Aristotle University of Thessaloniki, 54642 Thessaloniki, Greece

**Keywords:** kidney transplantation, donor-specific antibodies (DSAs), antibody-mediated

## Abstract

Kidney transplantation stands as the preferred treatment for end-stage kidney disease, significantly improving both the quality and longevity of life compared to dialysis. In recent years, the survival rates for patients and grafts have markedly increased thanks to innovative strategies in desensitization protocols for incompatible transplants and advancements in immunosuppressive therapies. For kidney transplant recipients, preventing allograft rejection is of paramount importance, necessitating the use of immunosuppressive medications. Regular follow-up appointments are essential, as monitoring the function of the kidney allograft is critical. Currently, established biomarkers such as serum creatinine, estimated Glomerular Filtration Rate (eGFR), proteinuria, and albuminuria are commonly employed to assess allograft function. However, these biomarkers have limitations, as elevated levels often indicate significant allograft damage only after it has occurred, thereby constraining treatment options and the potential for restoring graft function. Additionally, kidney biopsies, while considered the gold standard for diagnosing rejection, are invasive and carry associated risks. Consequently, the identification and development of new, sensitive, and specific biomarkers like dd-cfDNA, microRNAs (e.g., miR-21, miR-155), and sCD30 for allograft rejection are crucial. To tackle this challenge, intensive ongoing research employing cutting-edge technologies, including “omics” approaches, like genomic techniques, proteomics, or metabolomics, is uncovering a variety of promising new biomarkers.

## 1. Introduction

Kidney transplantation is the treatment of choice in patients with end stage chronic kidney disease in need of replacement therapy [1]. The relative risk of morbidity and mortality are increased in the first thirty days after transplantation. However, one year after transplantation, the risk is significantly lower, indicating a beneficial long-term effect when compared to wait-listed dialysis patients [2].

Furthermore, transplant recipients generally experience a better quality of life as they can return to normal activities and have fewer dietary restrictions in comparison to dialysis patients [3]. In addition, transplanted kidneys typically function better than kidneys that are artificially supported by dialysis, which leads to more normal physiological conditions and metabolic balance [3]. Therefore, access to transplantation remains a prominent public health priority [4,5,6,7].

One of the major problems in kidney transplantation is the risk of rejection, and as kidney transplant recipients require immunosuppressive medication for life, this can increase the risk of infections, malignancies, and other complications [8]. Moreover, there is often a shortage of suitable donor organs, leading to long waiting lists for patients in need of a transplant [9].

The length of time on hemodialysis before transplantation is associated with worse late transplant results [10]. Both preoperative and postoperative dialysis durations are critical factors influencing long-term kidney graft survival [10]. Moreover, as indicated by creatinine levels, residual renal function is also crucial for predicting graft survival [10].

Factors that influence the outcome and length of allograft survival include cardiovascular events; rejection episodes; infections; post-transplant neoplasia; donor quality; the occurrence of ischemia–reperfusion injury, which is associated with the duration of cold ischemia time; calcineurin inhibitor toxicity; the recurrence of the primary kidney disease; arterial stiffness; persistent arteriovenous access; mineral bone disease; fluctuations in immunosuppressive drug variability, which can lead to rejection or graft loss; hypomagnesemia; persistent inflammation; and metabolic acidosis [11].

Research has shown that patients undergoing pre-transplant PD may have a reduced risk of DGF compared to those on HD [12]. DGF is recognized as a factor associated with poor outcomes, including increased patient mortality and graft failure and also predisposes patients to the formation of DSA and other antibodies, which may adversely affect graft survival [12].

From an immunological perspective, PD is hypothesized to create a less pro-inflammatory environment than HD, potentially modulating the immune response during transplantation [12]. Chronic inflammation and oxidative stress, commonly associated with HD, are known to influence immune function and may contribute to antibody production [12]. This raises the possibility that patients on PD might exhibit a distinct immunological profile, potentially impacting the development of DSA [12].

Close monitoring with clinical and laboratory evaluation, using non-invasive biomarkers can indicate issues like subclinical acute rejection, a condition that can lead to chronic rejection and graft loss despite seemingly stable renal function, acute rejection, chronic allograft dysfunction, and ischemia–reperfusion injury, allowing for timely interventions [5,13] A molecule can be defined as biomarker when its levels can indicate and characterize normal biological processes, pathogenic processes, or responses to an exposure or intervention [2,6,7,14].

Traditional biomarkers, namely, serum creatinine and proteinuria, are imperfect and lag behind subclinical allograft injury [15]. Kidney allograft biopsies are still considered the gold standard for the diagnosis of allograft rejection [15]. While informative, they are invasive, with many possible complications like bleeding, hematomas, infections, damage to the surrounding tissues, and patient discomfort [15]. In addition, there is the inability to perform the procedure serially, resulting in variability in interpretation, and renal injury may not be detected until it has progressed significantly [15]. By the time histological changes are evident, substantial damage may have already occurred, limiting the effectiveness of therapeutic interventions [5,7]. It is therefore, vital to find more suitable biomarkers to monitor the viability of the allograft.

Furthermore, thrombotic microangiopathy (TMA) is a critical pathological entity in the context of kidney allograft rejection, presenting substantial diagnostic challenges due to its clinical and histopathological overlap with other conditions [11,16,17,18,19]. In renal transplantation, TMA may manifest as either recurrent (associated with a prior history of TMA in the native kidneys) or de novo (arising for the first time following transplantation) [20].

The diagnosis of TMA in kidney allografts primarily relies on the histopathological evaluation of renal biopsy specimens as the systemic manifestations of TMA are frequently absent [20]. Laboratory markers supporting the diagnosis of TMA include thrombocytopenia and microangiopathic hemolytic anemia, characterized by schistocytes on peripheral blood smears, decreased haptoglobin levels, and elevated lactate dehydrogenase levels [20]. However, these laboratory abnormalities are variably present and may not be consistently observed in all patients with TMA [20]. Additionally, the assessment of ADAMTS13 activity and the Coombs test are important diagnostic markers that should be evaluated in the context of suspected thrombotic microangiopathy (TMA) [20].

Another crucial consideration to keep in mind is the fact that the frequent thrombosis of vascular accesses can indicate a possible tendency to thrombosis, also known as a hypercoagulable state [21]. Patients with a history of recurrent vascular access occlusions may be considered at higher risk for thrombotic events [21,22].

The factors associated with an increased propensity for thrombosis, particularly in renal transplant recipients, encompass both clinical manifestations and specific laboratory markers, which include a personal or family history of venous thrombosis, diabetes mellitus, systemic lupus erythematosus (SLE), deficiencies in protein C, protein S, or antithrombin III, factor V Leiden (FVL) mutation (activated protein C resistance), prothrombin gene G20210A mutation, the presence of antibodies in β2-glycoprotein-1, lupus anticoagulant (LA) activity, the presence of antiphospholipid antibodies (APA), MTHFR C677T gene mutation, and hyperhomocystinemia [21]. Furthermore, donor and recipient factors that predispose to thrombosis are the presence of multiple vessels, atherosclerotic changes in the donor or recipient, pediatric donors/recipients, disseminated intravascular coagulation (DIC), the use of continuous ambulatory peritoneal dialysis (CAPD), repeat transplantation, and the use of cyclosporine or monoclonal antibodies [21]. These factors identify individuals at an elevated risk for thrombotic events, facilitating the implementation of targeted preventive measures [21].

In this article, we analyze only a snapshot of the field of biomarkers’ current state, so the following list is not exhaustive. The rapidly evolving nature of biomarker research means that promising new tests might not be included, which is crucial to note for responsible interpretations of the review.

## 2. Rejection: Definition and Types

Rejection refers to the transplantation of donor organs to non-HLA identical recipients, which introduces a stimulus into the recipient’s immune system, leading to immune attack and, eventually, to the damage of the allograft [16]. This is a complex process that involves various immune cells and mechanisms, and it is categorized in different ways depending on the specific cells and molecules involved, as well as the timing and characteristics of the damage [8,9], as it is discussed in more detail in the following paragraph of the article.

The current gold standard for diagnosing rejection is kidney biopsy [23]. Biopsy samples should contain at least 10 glomeruli and 2 small arteries [24]. While valuable, this procedure has many limitations, mainly in the context of its interpretation, since sampling errors can occur. Also, the molecular mechanism that preceded the injury may cause substantial damage to the allograft long before histological evidence of rejection, thus limiting available treatment possibilities [24].

According to the Banff criteria [16,17,18,19], types of rejection include active AMR, chronic active AMR, chronic inactive AMR, probable AMR, borderline acute TCMR, acute TCMR (IA, IB, IIA, IIb), and chronic TCMR (IA, IB, II). Furthermore, acute antibody-mediated rejection (AMR) can occur both early (<3 months) and late (>3 months) post-transplant [25]. In addition, AAMR is subclassified into three types according to the type of tissue injury: Type I, acute tubular necrosis (ATN)-like; type II, glomerular type, resembling thrombotic microangiopathy; and type III, vascular type with arterial inflammation [26].

Hyperacute rejection is a severe and immediate immune response that occurs within minutes to hours after transplantation [27]. It is characterized by widespread thrombosis of graft vessels due to pre-existing antibodies in the recipient’s blood targeting the donor organ [27]. It is triggered by the binding of high titers of anti-HLA antibodies to HLA type I molecules on the surface of the allograft’s endothelial cells, leading to direct tissue damage and the activation of the classical complement pathway, often accompanied by the immediate cyanosis of the graft, thrombosis of the blood vessels, and extensive tissue necrosis [24]. This process results in severe endothelial damage in the allogeneic transplant [24]. Specifically, the progressive release of heparan sulfate from the surface, mediated by the enzymatic cleavage of the protein core and glycosaminoglycan chains, leads to the loss of the endothelial barrier, which, in turn, results in thrombotic microangiopathy due to cell damage and consequent platelet aggregation and adhesion [23,28]. This type of rejection is rare today due to pre-transplant crossmatching and screening for donor-specific antibodies [24].

Delayed hyperacute or accelerated rejection (DHAR) is observed when there is an abrupt decline in urine output and graft tenderness occurring 3 to 14 days after transplantation [29]. This type of rejection is also associated with the presence of donor-specific anti- bodies, similar to hyperacute rejection, but manifests later in the post-transplant period [23]. It indicates an ongoing immune response against the graft, necessitating prompt evaluation and intervention. It is a severe type of acute humoral rejection that occurs within 2 weeks after ABO blood type-incompatible kidney transplantation [30]. Additionally, subclinical AMR is defined as immunohistological evidence of AMR in kidney transplant recipients with normal renal allograft function [31,32]. The term “acute vascular rejection” (AVR) is often ambiguously applied to all vascular lesions found during acute rejection. According to the Banff 2009 classification, AVR may fall into one of four categories: acute T cell-mediated rejection (ATMR) Type IIA, ATMR Type IIB, ATMR Type III, and acute antibody-mediated rejection (AAMR) Type III [16,18,19].

### 2.1. Risk Factors for Rejection

Blood transfusion, pregnancy, and a history of previous experience with solid-organ transplantation are the usual sensitizing events identified as risks for developing anti-human leukocyte antigen (HLA) antibodies [4].

Factors that impact de novo donor-specific antibody (DSA) development are medication nonadherence and excessive reductions in immunosuppressive agents, often to limit side effects [33]. Factors also include viral infections, especially cytomegalovirus (CMV) and Epstein–Barr virus; autoimmunity (AT1R); transplant nephrectomy; and HLA-DQ/-DR mismatches [34].

### 2.2. Types of Kidney Transplantation with Increased Immunological Risk

#### 2.2.1. DSAs—Incompatible Transplantation

The major histocompatibility complex (MHC), also known as the human leukocyte antigen (HLA) complex in humans, is a group of genes located on chromosome 6 that plays a crucial role in the immune system [35,36]. These genes encode proteins that present antigens to T cells, initiating an immune response. MHC class I proteins are expressed on the surface of nearly all nucleated cells, and they present fragments of intracellular proteins, such as viral proteins or tumor antigens, to CD8+ cytotoxic T lymphocytes, which in turn, activates them to destroy infected or cancerous cells. MHC class II proteins are primarily expressed on APCs, such as macrophages, dendritic cells, and B cells [36]. They present fragments of extracellular proteins, such as bacterial proteins or proteins from ingested pathogens, to CD4+ T helper cells [36,37]. This activation of T helper cells leads to the production of cytokines and antibodies, further amplifying the immune response.

MHC genes are highly polymorphic, meaning that they exist in many different versions or alleles within a population, and they are inherited in a codominant manner [36,38]. This incredible diversity ensures that pathogens will not evade easily the immune system. While polymorphism is beneficial for overall immune defense, it creates challenges in transplantation. This occurs because the recipient’s immune system may recognize the donor’s MHC as foreign, leading to allograft rejection [36,38].

HLA-incompatible kidney transplantation refers to a type of kidney transplant in which the donor and recipient have differences in their human leukocyte antigen (HLA) markers. This can lead to the production of antibodies against them, commonly termed donor-specific antibodies or DSAs [39,40]. HLA molecules are proteins found on the surface of cells that play a critical role in the immune system’s ability to recognize which cells belong to the body and which are foreign [34]. Non-HLA antibodies refer to antibodies that target antigens other than HLAs [33]. These antibodies can also contribute to graft dysfunction and rejection but are associated with different mechanisms and antigenic targets, such as anti-endothelial cell antibodies (AECAs) [33]. Non-HLA antibodies have been implicated in cases of acute dysfunction where anti-HLA DSAs are absent, indicating a distinct pathogenic role in transplantation [33].

##### HLA Epitopes Definition and Classification

HLA epitopes are the specific areas on HLA molecules where antibodies bind, defined by the tertiary conformation of amino acid sequences. These epitopes are characterized by the tertiary conformation of amino acid sequences, not just the primary sequence. On the other hand, HLA antigens are the molecules that induce an immune response and can be recognized by antibodies. The primary sequence of amino acids in a protein does not necessarily define an epitope, as epitopes are structurally defined areas.

The distinction between antigenicity (reactivity with anti-HLA antibody) and immunogenicity (capacity to induce anti-HLA antibody) is important [41,42].

The classification and definition of HLA epitopes, particularly through the lens of eplets (small configurations of amino acids on the HLA molecular surface, crucial for the formation of epitopes, both private and public), are critical for assessing immunological risk in organ transplantation [36]. Epitope mismatches significantly influence the development of anti-HLA antibodies, a process known as alloimmunization, which pose a risk for acute and chronic rejection, thus underscoring the importance of precise HLA matching to enhance transplant success and patient outcomes [36,43].

##### Characteristics of Donor-Specific Antibodies Associated with Pathogenicity

The most essential characteristic of DSAs associated with pathogenicity is the ability to activate the complement system [34,44].

DSAs that are subclass IgG1/IgG3- and C1q-activating have an 11-fold increased risk for ABMR and decreased 5-year graft survival.

Specific characteristics include specificity for HLA DQ mismatched antigens, a mean fluorescent intensity of more than 7000, C1q-activating capacity, and IgG1/IgG3 subclass [34,44].

#### 2.2.2. ABO-Incompatible Kidney Transplantation

Isoagglutinins (alloantibodies, isohemaglutinins) are naturally present and directed against the missing antigens from the individual’s RBCs [45,46]. They appear in the blood at early infancy (four to six months of age as a function of intestinal colonization with bacteria) [45,46]. Specifically, they are antibodies that occur against antigens not native to the host’s blood type [45,46]. In individuals with blood type O, antibodies to both A and B antigens are found, while those with blood type AB have no antibodies to A or B antigens [45,46]. These antibodies play a crucial role in determining compatibility for blood transfusions and organ transplants [45,46].

The ABO blood group system includes four categories, namely A, B, AB, and O, with different antigen expressions on various cells [39]. Blood group A has two subtypes, A1 and A2, with A1 being more immunogenic than A2 [39]. Recipients with blood type O have a higher risk of antibody-mediated rejection following ABO-incompatible transplantation [40]. The antigenic expression of the A carbohydrate antigen N acetylgalactosamine is reduced in the kidney cortex and endothelial surfaces of A2 donor kidneys, thereby making these kidneys inherently less antigenic to recipients with incompatible blood types [40]. ABO blood group incompatibility has been a significant barrier for living kidney donation due to the risk of antibody-mediated rejection [38]. Alloantibodies against missing antigens can lead to antibody-mediated graft damage and worse outcomes in recipients [38].

##### Complications of ABO-Incompatible Kidney Transplantation

ABOi transplants may face increased risks of viral infections (CMV, HSV, VZV, and BK virus), *P. jirovecii* pneumonia, and severe urinary tract infections [46,47]. In addition, the rate of posttransplant bleeding is higher in ABOi kidney transplantation recipients compared to ABO-compatible kidney transplant recipients [46,47,48,49].

Surgical complications after ABOi kidney transplantation are also increased, with a significantly higher number of lymphoceles requiring surgical revisions in ABOi patients compared to ABOc controls [46,47]. This is attributed to intensified immunosuppression and the removal of coagulation factors during the transplantation process [46,47].

#### 2.2.3. HLA-Incompatible Kidney Transplantation

HLA-incompatible kidney transplantation refers to a procedure where the donor and recipient have differences in their human leukocyte antigen (HLA) types. The presence of anti-HLA antibodies in the recipient’s blood, known as HLA sensitization, can result in antibody-mediated rejection and graft loss if not adequately managed [50].

##### Alloantibody Detection Tests

Complement-Dependent Cytotoxicity (CDC) Crossmatch: Traditional test detecting antibodies to HLA. It detects complement-fixing antibodies in the recipient’s serum that target donor lymphocytes. The test result is positive when there are enough antibodies to bind to the donor antigen and activate the complement cascade. CDC crossmatch can only detect complement-fixing antibodies and requires viable donor lymphocytes [51].

Flow Cytometry Crossmatch: Utilizes flow cytometry to detect antibodies against HLA. Flow cytometry crossmatch is a sensitive test that detects low-titer IgG DSAs, not observed in CDC crossmatch. It involves donor lymphocytes reacting with recipient serum using a flow cytometer and fluorochrome-conjugated antibodies. It is more sensitive than CDC crossmatch, aiding in the early detection of graft dysfunction and antibody-mediated rejection [52].

Solid-Phase Binding Assay (SPA): Measures DSAs using single antigen beads for immunologic risk stratification [44,53]. Solid-phase binding assays are used to detect HLA antibodies in organ transplant recipients [44,53]. This assay involves incubating antigen-coated microbeads with the recipient’s serum and then adding fluorescent-labeled anti-human IgG to detect the presence of anti-HLA antibodies [44,53]. It provides semiquantitave information through median fluorescence intensity (MFI) values [44,53]. It may present false positive or false negative results due to various factors in the recipient’s serum [44,53]. The desensitization treatment protocol involves initiating immunosuppression with PP/IVIG (number of plasmapheresis treatments combined with low dose IVIg) treatment, including FK506 and MMF [44,53]. Post-operatively, the treatment continues by maintaining FK506/MMF, prednisone taper, and ongoing PP/IVIG to achieve and maintain negative cytotoxic XM [44,53].

The Panel Reactive Antibody (PRA) Assay: An essential tool in transplant immunology, particularly for evaluating the sensitization status of patients awaiting kidney transplantation [45,46]. This assay measures the presence of antibodies against human leukocyte antigens (HLAs), which play a critical role in determining a patient’s eligibility for transplantation due to their impact on the risk of organ rejection [54,55]. Historically, the PRA assay relied on complement-dependent cytotoxicity (CDC) methods, which were limited by low sensitivity and an inability to distinguish between cytotoxic and non-cytotoxic antibodies [54,55].

The advent of solid-phase immunoassays, particularly single antigen bead (SAB) assays, has significantly enhanced the precision of HLA antibody detection and PRA calculation, since this innovation has led to the development of virtual PRA (vPRA), offering a more accurate assessment of sensitization [54,55].

##### Types of PRA Calculations

Total vPRA (vPRAt): Quantifies the patient’s lifetime sensitization history using a mean fluorescence intensity (MFI) cut-off of 1000. Includes antibodies resulting from all prior HLA sensitizing events [45].

Current vPRA (vPRAc): Focuses on antibodies detected within the past year, also using an MFI cut-off of 1000 [45].

Eplet-based vPRA (vPRAe): Provides a more granular analysis by evaluating eplets, the minimal immunogenic components of antigens [45]. Utilizes specialized software algorithms to enhance the identification and classification of HLA antibodies [45]. 

##### Impact on Transplantation Probability

Traditional vPRA methods (vPRAt and vPRAc) can disadvantage highly sensitized (HS) patients by reducing their likelihood of receiving compatible organ offers [45]. In contrast, eplet-based vPRA analysis has been shown to improve transplant probabilities by identifying acceptable donors more effectively and reducing waiting times for HS patients [45]. Studies indicate that the application of eplet analysis can lead to reclassification of many patients into lower vPRA intervals, thereby increasing access to compatible donors [45]. Desensitization treatment offers a significant survival benefit compared to other options like remaining on dialysis or waiting for a compatible kidney [48]. Combining desensitization with kidney-paired donation can be an effective strategy for transplanting sensitized patients and increasing transplant rates [48].

Specifically, in cases of highly sensitized individuals who have a positive crossmatch before kidney transplantation, desensitization strategies like plasmapheresis (PF) and intravenous immunoglobulin (IVIG) are commonly used to lower donor-specific antibody (DSA) levels, which is essential for improving the chances of transplant success [56,57]. After the desensitization process, it is important to reassess DSA levels to ensure they have sufficiently decreased and that the crossmatch is no longer positive [49]. This reassessment is particularly important in related living donor transplants, where compatibility between the donor and recipient plays a critical role in achieving successful transplant outcomes [56,57]. Plasmapheresis (PF) and low-dose intravenous immune globulin (IVIg) is part of the desensitization protocol before transplantation [49]. This approach aims to reduce donor-specific anti-HLA antibodies (DSA) to allow for transplantation to occur [49].

The above methods are described in Table 1.

## 3. Biomarkers of Graft Dysfunction

### 3.1. Biomarkers of Acute and Chronic Rejection Detected in Peripheral Blood Samples (Table 2)

#### 3.1.1. Biomarkers of Acute and Chronic Rejection Detected in Peripheral Blood Samples

Donor-derived Cell-free DNA (ddcfDNA): dd-cfDNA is a *sensitive* but not *specific* marker for allograft injury [51]. It refers to fragments of DNA originating from the transplanted kidney that are released into the recipient’s bloodstream. The fragments are typically 120–160 base pairs long and have a short half-life (about 30 min), meaning they are rapidly cleared by the liver and kidneys. Normally, only a small fraction (<0.2%) of circulating cfDNA is dd-cfDNA. However, in cases of allograft injury (including rejection), this fraction increases significantly [5,15,30,58]. A positive result (generally dd-cfDNA ≥1%) warrants further investigation in case of possible acute rejection or graft injury.

**Table 2 medicina-61-00262-t002:** Peripheral blood biomarkers for acute and chronic rejection in kidney transplantation patients. * Methods that are applied in clinical practice.

Biomarker	Method of Assessment	Pathogenic Characteristics/Function	Clinical Relevance
Donor-derived cell-free DNA (dd-cfDNA)	Quantitative PCR (qPCR), digital droplet PCR (ddPCR) *; next-generation sequencing (NGS) *; targeted NGS panels; fragmentation analysis (fragmentomics); methylation-specific methods; mass spectrometry-based methods; immunoassay-based methods; microarray-based techniques	Fragments of DNA from apoptotic donor cells, originating from the transplanted kidney, released into the recipient’s bloodstream, and initiating immune reactions.	Graft injury, AMR, and TCMR.
DSAs anti-HLA class I anti-HLA class II	Luminex single antigen bead (SAB) assay *; flow cytometry crossmatch *; complement-dependent cytotoxicity (CDC) assay *; C1q binding assay; IgG subclass analysis; ELISA-based methods; endothelial cell crossmatch	Bind to donor HLA class I and II molecules on endothelial cells and activate classical complement pathway.	Acute and chronic AMR.
Anti-MICA (MHC Class I-Related Chain A) Antibodies	Luminex-based assays *; ELISA *; flow cytometry; CDC assay; multiplex immunoassays; Western blot	Activate T lymphocytes and NK cells, leading to endothelial cell injury.	Acute and chronic AMR.
Anti-AT1R (Angiotensin II Type 1 Receptor) Antibodies	Luminex-based assays; ELISA *; cell-based assays; surface plasmon resonance (SPR); multiplex immunoassays	Immune and inflammatory responses, vasoconstriction, vascular injury, hypertension.	Acute and chronic AMR, chronic allograft dysfunction, fibrosis.
Anti-VEGF (Vascular Endothelial Growth Factor) Antibodies	ELISA *; surface plasmon resonance; radioimmunoassay; flow cytometry; functional neutralization assays; multiplex immunoassays; Western blot	Retard endothelial repair and angiogenesis.	Acute and chronic AMR, impaired vascular repair, chronic allograft dysfunction, fibrosis.
Non-HLA Autoantibodies	ELISA *; Luminex-based assays *; flow cytometry; Western blot; immunoprecipitation; functional neutralization assays; surface plasmon resonance (SPR); multiplex immunohistochemistry or immunofluorescence; next-generation sequencing (NGS)-based approaches	Chronic inflammation; classical complement activation.	Acute and chronic AMR.
Anti-C4d Antibodies	ELISA *; flow cytometry *; solid-phase assays (SPA); complement-dependent cytotoxicity (CDC) crossmatch; flow cytometric crossmatch	Ongoing complement activation.	Acute and chronic AMR.
Gene expression profiles (GEP) in peripheral blood	Commercially available GEP TestsAlloMap (CareDx) and TruGraf (Transplant Genomics) *; microarray analysis; quantitative real-time PCR; next-generation sequencing; multiplex PCR panels; digital PCR	T cell and B cell activation.	Early phases of acute rejection.
iATP levels	ELISA *; Western blot; flow cytometry; RIA; immunoprecipitation; solid-phase assays (SPA)	Ischemic injury; mitochondrial dysfunction; inflammation; tubular injury.	Increased levels suggest acute rejection; reduced levels suggest infection.
Donor-specific IFN-gamma-producing lymphocytes	ELISPOT assay *; ELISA *; lymphocyte transformation test (LTT) *; flow cytometry; cytotoxicity assays; multiplex bead assays	immune activation; immunologic memory; vascular damage; fibrosis.	Increased levels predict acute rejection.
sCD30	ELISA *; Western blot; flow cytometry; immunoprecipitation assay; cytotoxicity assays; multiplex bead assays	A glycoprotein expressed on human CD4+ and CD8+ T cells that secrete Th2-type cytokines.	Reflects recipients who may generate an alloimmune response against a grafted kidney.
Donor-specific IFN-gamma-producing lymphocytes	ELISA *; LTT *; IFN-gamma enzyme-linked immunospot (ELISPOT) assay *; flow cytometry; cytotoxicity assays; multiplex bead assay; solid phase assays	Direct cytotoxicity, endothelial injury, activation of macrophages and dendritic cells, expression of MHC class I and II.	Reflect immunologic memory and correlate with the risk of post-transplant rejection episodes.
Torque Tenovirus (TTV)	ELISA *; Western blot; immunofluorescence assay; PCR for TTV DNA; immunoprecipitation; multiplex bead assay	TTV viral load in peripheral blood might reflect the intensity of host immunosuppression.	Active immune responses.
Plasma endothelial microparticles	ELISA *; flow cytometry *; Luminex-based assays; Western blotting; immunoelectron microscopy; ELISPOT assays	Endothelial dysfunction.	Early phases of AMR.
Titin; lipopolysaccharide-binding protein; peptidase inhibitor 16; complement factor D; mannose-binding lectin; protein Z-dependent protease; 2 -microglobulin	ELISA *; Luminex-based assays *; Western blot; flow cytometry	Complement activation; dysregulation of coagulation; chronic inflammation leading to fibrosis.	Proteins increased in AR.
Kininogen-1; afamin; serine protease inhibitor; phosphatidylcholine-sterol acyltransferase; and sex hormone-binding globulin	ELISA *; Luminex-based assays *; Western blot; flow cytometry	Reduce oxidative stress; promote anti-inflammatory and vasoprotective effects; reduce inflammation.	Proteins decreased in AR.
Mitochondrial DNA (mtDNA)	ELISA *; immunofluorescence assay (IFA) *; Luminex-based assays; Western blot; immunoprecipitation assay; RIA	Direct cellular injury; cytokine production; vascular injury; fibrosis.	AR; vascular injury; and chronic graft dysfunction.
Anti-LG3 (Perlecan Fragment) Antibodies	ELISA *; Luminex-based assays *; Western blot; immunofluorescence assay (IFA); flow cytometry; functional assays	Amplify complement activation.	AR; thrombotic microangiopathy; chronic graft dysfunction; microvascular inflammation.
Anti-Endothelial Cell Antibodies (AECA)	Flow Cytometry *; ELISA *; immunoprecipitation assay; Western blot; Luminex-based assays	Endothelial activation.	AR; chronic graft dysfunction; microvascular inflammation.
MicroRNAs (e.g.; miR-21; miR-155)	ELISA *; RNA immunoprecipitation *; Western blot; RT-qPCR; multiplex assay	Non-coding RNAs regulating gene expression in immune and inflammatory pathways.	Altered expression patterns correlate with acute rejection.
TIM-3	ELISA *; flow cytometry *; multiplex bead assays *; Western blot	Immune checkpoint protein regulating T cell activation. Elevated levels indicate failed immune regulation.	Increased levels predict acute rejection and immune activation.
Perforin	ELISA *; flow cytometry *; multiplex bead assays *; Western blot	Protein secreted by T cells and NK cells; cytotoxic activity	Increased levels indicate acute rejection.
Granzyme B	ELISA *; flow cytometry *; multiplex bead assays *; Western blot	Protein secreted by T cells and NK cells; cytotoxic activity and apoptosis.	Increased levels are associated with T cell-mediated acute rejection.
CXCL9 and CXCL10 (Monokine induced by gamma-interferon and Interferon-inducible protein-10)	ELISA *; multiplex bead assays *; flow cytometry; Western blot; RIA	Stimulate T cell recruitment to the kidney graft during rejection.	Increased levels in urine predict acute rejection.
Pro-inflammatory Cytokines (e.g.; IL-6 IL-2; IL-17; TNF-a)	ELISA *; flow cytometry *; multiplex bead assays *; Western blot	Pro-inflammatory cytokines (e.g., IL-6 IL-2, IL-17, TNF-a).	Active immune responses, acute rejection.
Autophagy-related proteins (LC3 (microtubule-associated protein light chain 3), Beclin-1, and p62).	Assays for monitoring autophagy (multiple assays to monitor autophagy flux, which is a measure of autophagic degradation activity).	Cellular materials are encapsulated in double-membrane structures known as autophagosomes, fusing with lysosomes to degrade the contents.	Indicators of autophagic activity within renal cells and may provide insights into their potential involvement in graft injury or rejection.

This method shows promise and greater diagnostic value in identifying antibody-mediated rejection (AMR) but struggles with T cell-mediated rejection (TCMR). Studies show significantly higher levels of dd-cfDNA in ABMR compared to TCMR, even in DSA-negative cases [30,58].

Retrospective studies suggest that dd-cfDNA might detect allograft injury months before a clinical diagnosis of rejection (both AMR and TCMR). This suggests the potential for the early detection of problems, allowing for preemptive intervention to prevent progression to overt rejection. A significant decline in dd-cfDNA levels after the initiation of antirejection therapy is associated with a positive treatment response. This suggests that monitoring dd-cfDNA might help identify patients who are responding well to treatment versus those who might require alternative therapies or further investigation (e.g., a repeat biopsy) [15,59].

Donor-Specific Antibodies (DSAs): The introduction of solid phase immunoassay technologies has recently allowed a greater sensitivity in the detection and characterization of human leukocyte antigen (HLA) antibodies compared to traditional complement-dependent lymphocytotoxicity (CDC) assays [44,60]. Preformed DSA is defined as an antibody detected prior to transplant or as a new DSA that develops in the first 2 weeks to 3 months post-transplant. De novo DSA is defined as the onset of a new DSA occurring more than 3 months post-transplant [54]. The recommended methods include using solid-phase assays that encompass all major HLA class I and II loci, with mean fluorescence intensity (MFI) thresholds set at 1000 to 1500 MFI as universal cutoff values for positivity [54]. Anti-HLA class I antibodies target HLA class I molecules, which are present on nearly all nucleated cells and play a critical role in presenting endogenous antigens to CD8+ cytotoxic T cells. Anti-HLA class II antibodies target HLA class II molecules, which are primarily expressed on professional antigen-presenting cells (such as dendritic cells, macrophages, and B cells) and are important for presenting exogenous antigens to CD4+ helper T cells. These include HLA-A, HLA-B, HLA-C, HLA-DRB1, HLA-DRB3/4/5, HLA-DQA1/DQB1, and HLA-DPA1/DPB1. The above-mentioned antibodies can have varying attributes such as mean fluorescence intensity (MFI), complement-fixing ability (C1q-fixing), and subclasses (e.g., IgG subclasses) [54]. They are responsible for acute and chronic AMR.

DSAs MF, especially those with complement-fixing IgG subclasses, are more strongly associated with C4d deposition and potential AMR. This highlights the clinical value of assessing both the quantity and quality (complement-fixing capacity) of DSAs [54].

Antibodies against Major-Histocompatibility-Complex (MHC) Class I-Related Chain A (MICA) and Chain B (MICB): MICA and MICB antigens in humans consist of a family of polymorphic genes that play a major role in immune responses. They can activate the NKG2D receptor, a member of the killer cell lectin-like receptor complex, expressed on memory-effector T cells or natural killer (NK) cells, providing a signal to help activate their effector cytolytic response [55]. MICA is expressed in various cell types, including keratinocytes, endothelial cells, fibroblasts, and monocytes. Elevated MICA levels on grafts may trigger anti-MICA antibody production, contributing to graft rejection [55]. Studies have shown that they are strongly associated with increased rates of AMR and reduced graft survival [61].

Anti-Angiotensin II Type 1 Receptor (AT1R) antibodies: AT1Rs are a type of G-protein-coupled receptor that mediates the physiological actions of angiotensin II, a peptide hormone involved in various regulatory functions in the body. These receptors are widely distributed in various tissues, including liver, lungs, vasculature, brain, heart, kidneys, adrenal glands, and placenta. Anti-AT1R antibodies are associated with AMR and have been shown to negatively impact graft survival in kidney transplant recipients. When these antibodies bind to their receptor, they can activate signaling pathways that contribute to inflammatory responses and vascular injury. Additionally, they can sensitize recipients to other antigens, particularly HLA [56].

Anti-Vascular Endothelial Growth Factor (VEGF) Antibodies: Anti-VEGF antibodies may be associated with renal allograft rejection through their impact on the VEGF signaling pathways, whose controlled function is crucial for maintaining the integrity of the glomerular filtration barrier. In the context of kidney transplantation, the use of anti-VEGF therapy can lead to glomerular endothelial injury and contribute to proteinuria, which can be indicative of allograft dysfunction [57].

Non-HLA Autoantibodies: Non-HLA antibodies are antibodies directed against autoantigens that are not a highly polymorphic HLA antigen [58]. They are significant because their development is associated with rejection and decreased long-term graft survival, contributing to antibody-mediated acute and chronic rejection [58]. Examples of these autoantibodies include (1) anti-endothelial cell antibodies (AECAs), which are implicated in hyperacute rejection and accelerated AMR; (2) anti-AT1R antibodies, which are described in detail above; (3) antibodies against perlecan (specifically the LG3 fragment), which are associated with acute vascular rejection; and (4) antibodies against Vimentin, collagen IV, K-alpha 1 tubulin and fibronectin, associated with transplant glomerulopathy in renal transplant recipients [58].

Antibodies Against C4d and Allo-Antigens: C4d is a degradation product of the activated complement factor C4. It is accumulated after the binding of antibodies to specific target molecules, which in turn leads to complement activation and subsequent deposition of C4d at sites of injury. Therefore, C4d deposition is indicative of AMR. Evidence suggests that a significant percentage of patients with C4d-positive biopsies have detectable antibodies, particularly donor-specific antibodies directed against MHC class I and/or class II antigens [59]. The presence of C4d in urine or blood indicates AMR. The quantification of plasma C4d+ microvesicles, which are membrane-bound vesicles released from the cell surface following injury, can provide information about the presence and severity of AMR and may help in monitoring treatment response [60].

Gene Expression Profiles (GEP) in Peripheral Blood: The simultaneous measurement of thousands of genes by applying microarray technology in the peripheral blood of kidney transplant recipients is a non-invasive technique that can be used to detect early, but not yet clinically acute, rejection. It provides a binary result: Transplant “excellent” (TX), which indicates a low risk of subclinical rejection, and non-TX, which suggests a higher risk of subclinical rejection, warranting further investigation, including graft biopsy. It demonstrates a high negative prognostic value, indicating its effectiveness in ruling out subAR and potentially reducing unnecessary biopsies. However, it has a lower positive prognostic value, limiting its use in definitively diagnosing subAR [7,9]. Despite its advantages, it has been withdrawn from the market.

Intracellular Adenosine Triphosphate (iATP). iATP levels are measured by immune cell function assay in CD4+ T cells after in vitro stimulation. The original hypothesis was that this assay could assess the overall immune function of kidney transplant recipients, potentially predicting the risk of both infection and rejection. Early studies suggested that low iATP levels (<225 ng/mL) indicated under-reactive immune functions, increasing the risk of infectious complications, whereas high iATP levels (>525 ng/mL), suggested an overactive immune response, potentially increasing the risk of rejection [15].

Donor-Specific IFN-Gamma-Producing Lymphocytes: They are considered to reflect immunologic memory and correlate with the risk of post-transplant rejection episodes. This measurement is used as part of the IFN-gamma enzyme-linked immunospot (ELISPOT) assay [5]. 

Soluble CD30 (sCD30): sCD30 is a glycoprotein expressed on human CD4+ and CD8+ T cells that secrete Th2-type cytokines [62]. It reflects recipients who may generate an alloimmune response against a grafted kidney, predicting a poor graft outcome [63].

Torque Tenovirus (TTV): This is a ubiquitous human DNA virus. Research exploring its potential role as a biomarker in kidney transplantation is relatively recent and the results are mixed and not conclusive. The initial hypothesis was based on the idea that the TTV viral load in peripheral blood might reflect the intensity of host immunosuppression. However, the sensitivity and specificity of TTV as a biomarker for rejection or infection were modest in early studies [15].

Plasma Endothelial Microparticles: These are being investigated as promising markers to assess endothelial dysfunction in kidney transplantation and could be used as early diagnostic biomarkers of AMR. These microparticles (plasma-derived microparticles and endothelium cell-derived microparticles) play vital roles in intercellular communication, inflammation, and coagulation [64].

Proteins Increased in Biopsy-Confirmed Acute Rejection (bcAR): Titin, lipopolysaccharide-binding protein, peptidase inhibitor 16, complement factor D, mannose-binding lectin, protein Z-dependent protease, and 2-microglobulin were found to be increased in patients with biopsy-confirmed acute rejection (BCAR) [65].

Proteins Decreased in bcAR: Kininogen-1, afamin, serine protease inhibitor, phosphatidylcholine-sterol acyltransferase, and sex hormone-binding globulin were identified as proteins that were decreased in patients with BCAR [65].

Mitochondrial DNA (mtDNA): Elevated levels of mtDNA in donor plasma predict delayed graft function (DGF), particularly in donors after cardiac death (DCD). This suggests that mitochondrial damage reflects overall kidney quality. It is predictive and non-invasive, but needs further validation across diverse donor populations [15].

Autophagy-related proteins commonly investigated in scientific studies include LC3 (microtubule-associated protein light chain 3), Beclin-1, and p62 [66]. The expression levels of these proteins serve as indicators of autophagic activity within renal cells and may provide insights into their potential involvement in graft injury or rejection [66].

#### 3.1.2. Urinary Biomarkers

They can provide real-time information about kidney function. However, they can degrade rapidly after collection and standardization is necessary [5,15,59,66,67,68,69,70,71,72,73,74,75].

Messenger RNA (mRNA): Studies have evaluated specific mRNA transcripts as indicators of AR. One study validated a 3-gene signature for detecting T cell-mediated rejection. However, challenges in preserving urine mRNA for accurate analysis need to be addressed before widespread use.

Other studies have found that recipients experiencing acute rejection showed significantly higher levels of soluble T cell immunoglobulin and mucin domain-containing protein 3 (sTIM-3) and perforin mRNA than those without rejection. Furthermore, sTIM3 can also effectively predict steroid-resistant, as well as the response to anti-rejection therapy. Perforin can additionally be used for predicting chronic allograft dysfunction [76,77].

Fibrinogen Alpha, Beta, and Gamma Chain (FGA FGB, and FGG): These proteins are components of fibrinogen, which is involved in blood clotting and can be useful in detecting ischemia–reperfusion injury (IRI) and acute or chronic allograft rejection. They can also serve as therapeutic targets to prevent thrombosis or rejection [71].

Neutrophil Gelatinase-Associated Lipocalin (NGAL) and Kidney Injury Molecule-1 (KIM-1): These two molecules, as markers of tubular epithelial cell injury, may be excreted in the urine of kidney transplant recipients, indicating either acute rejection episodes or any other forms of tubular epithelial cell injury [72].

Keratins and Histones: Keratins are structural proteins of epithelial cells, and histones are involved in DNA packaging and gene regulation. Their urinary levels are both increased in cases of ischemia–reperfusion injury [67].

Proteins and Chemokines: Proteins and chemokines, such as IL-18, TNF-a, CXCL9, and CXCL10 are involved in inflammatory responses, and specifically stimulate T cell recruitment to the kidney graft during rejection. Increased urinary excretion predict acute rejection [78,79]. Granzyme B is another protein secreted by T cells and NK cells and is indicative of their cytotoxic activity and apoptosis. Increased levels are associated with chronic active antibody-mediated acute rejection [77].

Extracellular Vesicles (EVs): These encompass exosomes (small vesicles released from cells), microvesicles (shed from the plasma membrane), and apoptotic bodies (released during cell death). They are involved in intercellular communication, influencing both local (paracrine) and systemic (endocrine) responses. They can be used to detect acute cellular rejection. Their presence and composition in urine or plasma can represent status of graft function. They are also associated with chronic rejection processes and can provide insights into the ongoing health of the transplanted kidney. In addition, they can help predict DGF, allowing for timely interventions. The analysis of urinary EVs has a proved a diagnostic and prognostic potential [75,78,79,80,81,82,83,84,85].

Exosomes: These extracellular vesicles contain various molecules reflective of kidney function and status. Research into using urinary exosomes as biomarkers is ongoing and standardization is still needed [80].

“Q-score”: This composite biomarker included urinary cfDNA, methylated cfDNA, CXCL10, creatinine, clusterin, and total protein, showing potential to predict rejection. However, this assay (Q-Sant) has been withdrawn due to limitations [80].

Fifteen-gene mRNA Signature Derived from Urinary EVs (ExoTRU (Bio-Techne): This process can assess kidney transplant health and identify acute rejection (AR), and it includes the following genes: CXCL11, CD74, IL32, STAT1, CXCL14, SERPINA1, B2M, C3, PYCARD, BMP7, TBP, NAMPT, IFNGR1, IRAK2, and IL18BP [81].

Apolipoprotein A1 (APOA1): APOA1 in plasma was identified as a potential biomarker for acute cellular rejection in kidney transplant recipients utilizing surface-enhanced laser desorption/ionization time-of-flight mass spectrometry (SELDI-TOF MS). Specifically, decreased levels of APOA1 are associated with acute cellular renal allograft rejection [82].

C-Terminal Fragment of α-1 Antichymotrypsin: This protein comprises a plasma protein that can be identified by utilizing surface-enhanced laser desorption/ionization time-of-flight mass spectrometry (SELDI-TOF MS). Decreased levels of this protein are indicative of acute cellular rejection [82].

Transthyretin (TTR): Studies have shown that TTR levels were significantly higher in the urine extracellular vesicles (EVs) of kidney transplant recipients with chronic active antibody-mediated rejection [67].

Polymeric Immunoglobulin Receptor (PIGR): Studies have shown that this biomarker has a 76.2% sensitivity in differentiating between CAMR and other groups [80].

Hemopexin (HPX): This has been identified as a potential biomarker for acute T cell-mediated rejection (TCMR) in kidney transplant recipients (KTRs) [80].

Zinc-alpha-2-glycoprotein (AZGP1): This protien was significantly increased in CAMR patients compared to the control group [80].

Ceruloplasmin (CP): This is a potential biomarker, capable of differentiating CAMR from other conditions [80].

C4d: Increased urinary excretion of C4d indicates complement activation and can be used as a non-invasive marker of AMR [80].

Complement C5a: High urinary levels in donors predict DGF. This points to the role of complement activation in ischemia–reperfusion injury (IRI). It is non-invasive, but may not be specific to DGF [67]. 

### 3.2. Biomarkers in Renal Transplantation Ischemia–Reperfusion Injury and Delayed Graft Function

The following biomarkers were discovered through combined experimental studies and omics approaches, including transcriptome and proteome analysis. It is anticipated that a combination of these biomarkers will be effective in identifying individuals at risk for IRI and DGF, which could allow for prophylactic measures to be taken, potentially leading to reduced rates and IRI severity and increased graft longevity. Many of the following biomarkers are indicative of tubular or vascular damage and can predict the incidence and severity of IRI and DGF before transplantation [5,83].

Urinary Kidney Injury Molecule-1 (KIM-1): This molecule is identified as a sensitive biomarker for the early detection of kidney tubular injury. Its levels correlate with the severity of acute renal failure [5,83].

Keratinocyte-Derived Chemokine: This chemokine serves as an early biomarker for ischemic acute kidney injury, reflecting its involvement in the inflammatory response [5,86,87,88,89].

Annexin A2 and S100A6: These are calcium-binding proteins that function as sensors of tubular injury and aid in recovery during acute renal failure [5,89].

Cystine Rich Protein 61 (CYR61, CCN1): These proteins are detected early in urine after renal ischemia–reperfusion injury, suggesting early kidney stress or damage [5,89].

S100B: This protein’s release patterns differ during ischemia and reperfusion processes in organs like the liver, gut, and kidney, indicating tissue damage responses [5,83].

Serum Cystatin C: Known for the early detection of acute renal failure, this biomarker reflects changes in kidney function before other indicators [5,89].

Neutrophil Gelatinase-Associated Lipocalin (NGAL): An essential biomarker [5,83] for acute kidney injury, notably after cardiac surgery [5,89].

Netrin-1: Its expression is linked to acute kidney injuries [5,83].

Endoglin: This protein plays a regulatory role in renal ischemia–reperfusion injury, making it significant for kidney function assessments [5,89].

Lipocalin 2: This marker is noted for its role in detecting kidney injuries, with increased levels signaling acute damage [5,89].

Complement Component 3 (C3): The local extravascular pool of C3 is influential in postischemic acute kidney failure, denoting an immune response [5,89].

Fatty Acid Binding Protein: This protein serves as a marker for renal injury and is monitored for insights into ischemic incidents impacting kidney tissues [5,89].

Activating Transcription Factor 3 (ATF3): Protects against renal ischemia–reperfusion injury, acting as a resilience factor during such incidences [5,89].

Cyclin-Dependent Kinase Inhibitor: This indicator aids in the understanding of cell cycle regulation after renal ischemia [5,89].

Uromodulin (UMOD): Associated with tubular function and injury [5,89].

Interleukin-18 (IL-18): Associated with inflammatory response and is known to predict DGF [5].

Other biomarkers in this category include ACTA2 (actin, alpha 2, smooth muscle, aorta), LGALS3 (lectin, galactoside-binding, soluble, 3), SAT1 (spermidine/spermine N1-acetyltransferase 1), HAVCR1 (hepatitis A virus cellular receptor 1), CXCL1 (chemokine (C-X-C motif) ligand 1), ANXA2 (annexin A2), S100A6 (S100 calcium-binding protein A6), CYR61 (cysteine-rich angiogenic inducer 61), S100B (S100 calcium-binding protein B), AMBP (alpha-1-microglobulin/bikunin precursor), LCN2 (Lipocalin 2), C3 (complement component 3), FABP1 (fatty acid-binding protein 1, liver), ATF3 (activating transcription factor 3), GUCY2G (guanylate cyclase 2G), and BID (BH3-interacting domain death agonist) [5].

### 3.3. Other Types of Biomarkers [78]

Novel biomarkers identified by newer techniques are described in Table 3.

#### 3.3.1. Genes Overexpressed in the Common Rejection Module

The following genes were discovered through genomic studies and analyses involving kidney transplant patients. Various studies utilized techniques like microarray analysis, qPCR, and computational gene expression scoring to identify genes associated with acute rejection (AR) and chronic allograft dysfunction (CAD) [5].BASP1: Brain-abundant membrane-attached signal protein 1, located on 5p15.1.CD6: CD6 molecule, found on 11q12.2.CXCL10: C-X-C Motif chemokine ligand 10, situated on 4q21.1.CXCL9: C-X-C Motif chemokine ligand 9, located on 4q21.1.INPP5D: Inositol polyphosphate-5-phosphatase D gene, present on 2q37.1.ISG20: Interferon stimulated exonuclease gene 20, found on 15q26.1.LCK: LCK protooncogene, a SRC family tyrosine kinase, situated on 1p35.2.NKG7: Natural killer cell granule protein 7, located on 19q13.41.PSMB9: Proteasome subunit beta 9 gene, found on 6p21.32.RUNX3: Runt-related transcription factor 3, situated on 1p36.11.TAP1: Transporter 1, ATP binding cassette subfamily B member, located on 6p21.32. 

#### 3.3.2. Transcriptomic Biomarkers

Transcriptomic biomarkers are identified through gene expression profiling using microarray and next-generation sequencing technologies, often on renal biopsy samples. The goal is to find gene signatures associated with fibrosis, i-IFTA (interstitial fibrosis and tubular atrophy), and chronic rejection (ABMR—antibody-mediated rejection. Many genes are mentioned in groups, making individual analysis difficult without the original data. One study identified eleven genes associated with acute rejection and seven that were predictive of progressive i-IFTA at 24 months post-transplant. Another study used a four-gene model (vimentin, NKCC2, E-cadherin, and 18S rRNA) in urine to diagnose i-IFTA. Markers associated with antibody-mediated rejection (ABMR) were predominantly detected in endothelial cell genes, highlighting the critical role of endothelium in the pathogenesis of ABMR [79].

#### 3.3.3. Epigenetic Biomarkers

These reflect changes in gene expression *without* altering the DNA sequence. They include DNA methylation, microRNA interactions, and histone modifications. Foxp3 DNA demethylation correlate positively with the number of intragraft Foxp3-expressing T cells (important regulatory T cells), indicating better graft outcomes. PD1 DNA methylation in memory CD8+ T cells shows increased levels in rejection. Several microRNAs (miRs) are highlighted, such as miR-21, miR-200b, miR-150, miR-192, miR-200b, miR-423-3p, miR-145-5p, miR-148a, miR-142-3p, miR-204, and miR-211. Their expression levels in urine or plasma are correlated with various aspects of CKTR, such as IFTA or CAD (chronic allograft dysfunction) [84].

#### 3.3.4. Proteomic Biomarkers

These biomarkers are proteins identified using various high-throughput proteomic techniques (mass-spectrometry (MS), liquid chromatography (LC), iTRAQ, etc.) in urine or blood samples. Proteins such as β2-microglobulin (B2M) have been associated with chronic allograft disease and are indicative of renal function impairment [85]. 

#### 3.3.5. Metabolomic Biomarkers

Metabolomic biomarkers are metabolites, i.e., small molecules involved in cellular processes. Specific metabolites, including NAD, 1-MN, cholesterol sulfate, GABA, nicotinic acid, NADPH, proline, and spermidine, can be used to predict TCMR or overall chronic allograft nephropathy through urine or blood samples. Furthermore, S-adenosyl methionine and S-adenosyl homocysteine, as well as metabolites from the arachidonic acid bioactive lipid pathway (like 18-HEPE and 12-HETE), may serve as predictive markers of kidney transplant rejection [86]. 

#### 3.3.6. Cellular Biomarkers

These focus on specific immune cell populations and their characteristics, using techniques like flow cytometry. For example, alloreactive CD8+ T cells, particularly effector memory T cells (TEMRA and EM). CD154+ T-cytotoxic memory cells are noted as being associated with rejection risk, while the ratio of T follicular helper cells and T follicular regulatory cells (Tfh/Tfr) is linked to CAD. Macrophages and NK cells’ roles are also mentioned, but specific subsets have not yet been defined as robust biomarkers [86]. 

### 3.4. Specificity and Sensitivity of Biomarkers

Presented below is a representative list of some of the most commonly utilized biomarkers, accompanied by a description of their specificity and sensitivity.

Donor-Derived Cell-Free DNA (dd-cfDNA): This biomarker is noted for its sensitivity but is not highly specific for allograft injury. High levels may indicate acute rejection or graft injury, with a positive result generally suggesting further investigation [87].Donor-Specific Antibodies (DSAs): The introduction of solid-phase immunoassay technologies has improved the sensitivity in the detection of DSAs compared to traditional methods. DSAs, especially those that are complement-fixing, are strongly associated with acute and chronic antibody-mediated rejection, underscoring their clinical relevance [15]. Flow cytometry cross-matching exhibited a specificity of 95% but a sensitivity of only 35% for predicting antibody-mediated rejection (AMR). This indicates that while the test is reliable in identifying patients who will not experience AMR (high specificity), it fails to detect a significant portion of those who will experience AMR (low sensitivity) [88].Non-HLA Antibodies: These antibodies, which target autoantigens not within the highly polymorphic HLA antigens, are significant as their development is linked to rejection and diminished long-term graft survival. Non-HLA antibodies include anti-endothelial cell antibodies, anti-angiotensin II Type 1 receptor antibodies, and others [15].

Research has shown that measuring the levels of four antibodies—MIG (CXCL9), ITAC (CXCL11), IFN-g, and glial-derived neurotrophic factor (GDNF)—could help clinicians predict the development of CAI with over 80% sensitivity (this means that the test is effective at detecting the presence of CAI in patients who have it) and 100% specificity (meaning that the test is highly effective at confirming that a patient does not have CAI when they actually do not). While non-HLA antibodies can be detected, their clinical relevance in predicting rejection may be limited as they may not lead to significant detrimental effects on the allograft [88].In conclusion, non-HLA antibodies may provide some information regarding the immune status of a transplant recipient, but their predictive value for kidney allograft rejection appears limited when using standard cross-matching techniques. The high specificity of the FC cross-match is promising, but its low sensitivity necessitates caution, as many patients who experience AMR may not be identified through current testing [88].Sensitivity and specificity of antibodies to the angiotensin II type 1 receptor (AT1R) in the context of kidney allograft rejection using enzyme-linked immunosorbent assay (ELISA) has 96% specificity, which means that there is a low chance of false positives (indicating that patients identified as negative for anti-AT1R antibodies are very likely truly negative), and 88% sensitivity, which indicates that the test is fairly good at identifying individuals who do have high levels of anti-AT1R antibodies, which are associated with antibody-mediated rejection [89].

### 3.5. Classification of Biomarkers According to Their Clinical Applications

Table 4 includes only a few examples of the many existing biomarkers, and it is important to note that a single biomarker can fall into multiple categories based on its specific use. 

## 4. Advantages and Limitations 

Non-invasive biomarkers are emerging as vital tools in kidney transplantation, offering numerous advantages for monitoring graft health and enhancing patient outcomes [15,52]. Among their primary benefits is the ability to detect allograft injury or rejection at an early stage [15,59]. Biomarkers like donor-derived cell-free DNA (dd-cfDNA) allow for timely intervention, potentially preventing graft loss [15]. Additionally, non-invasive approaches, such as urine-based biomarkers, minimize the need for kidney biopsies, thereby reducing associated risks like bleeding and infection [15]. These biomarkers also enhance the capacity for graft health monitoring and treatment response evaluation, enabling personalized immunosuppressive therapies [5,8,32,90]. The diversity of biomarkers, including gene expression profiles, protein levels, and specific antibodies, provides a comprehensive understanding of both acute and chronic rejection mechanisms [5,8,33,34].

Despite their advantages, non-invasive biomarkers face certain limitations [9]. Some, like dd-cfDNA, exhibit high sensitivity but lack specificity, as elevated levels may arise from causes other than allograft injury, such as infections [9]. Moreover, interpreting biomarker levels can be complex, as various factors, including comorbidities, can influence results [7,14,15]. The utility of biomarkers also varies; for instance, dd-cfDNA is more effective in detecting antibody-mediated rejection (AMR) than T cell-mediated rejection (TCMR) [15]. Additionally, the high cost and limited accessibility of advanced biomarker testing present challenges, particularly in resource-limited settings [14,58].

Further research and validation are essential for overcoming these challenges [5,30,59,88]. Standardizing biomarker assays is critical to ensure reliable and consistent results across laboratories [5,30,59,88]. Longitudinal studies are needed to establish the predictive value of biomarkers over time, especially across diverse populations and clinical contexts [5,30,59,88]. Comparative studies are also necessary to assess the effectiveness of these newer biomarkers against established measures, such as serum creatinine levels and kidney biopsies [5,30,59,88]. Moreover, the development of clear guidelines on incorporating biomarkers into routine clinical practice, including recommendations on testing frequency and timing, is imperative [5,30,59,88].

To sum up, non-invasive biomarkers represent a promising advancement in kidney transplantation, offering significant potential to improve patient outcomes through early detection, enhanced monitoring, and reduced dependence on invasive procedures. However, addressing challenges related to specificity, cost, interpretation, and integration into clinical practice will be essential to fully realize their benefits.

## 5. Conclusions

The article emphasizes the essential function of non-invasive biomarkers for the early identification and monitoring of rejection and graft dysfunction in kidney transplant patients. Conventional techniques, such as measuring serum creatinine levels and conducting kidney biopsies, are fraught with significant drawbacks, including their invasive nature and the potential for the delayed detection of allograft damage. Progress in biomarker research presents promising alternatives that can enable prompt interventions and enhance patient outcomes. A variety of biomarkers—including donor-derived cell-free DNA, specific antibodies, and gene expression profiles—offer valuable insights into both acute and chronic rejection, thereby aiding in the customization of immunosuppressive treatments and improving graft survival rates. Donor-specific antibodies (DSAs), particularly those that bind complement, are considered important in diagnosing kidney allograft rejection. Studies have shown that the presence of complement-binding DSAs is associated with an increased risk of graft failure, making them relevant biomarkers in the context of antibody-mediated rejection (ABMR) [91]. Additionally, dd-cfDNA (donor-derived cell-free DNA) is emerging as a promising biomarker for the diagnosis of allograft rejection. It provides a non-invasive means to detect injury to the graft, and its levels can correlate with the severity of rejection [91].

As research advances, incorporating these biomarkers into clinical settings will be crucial for optimizing kidney transplant care and tackling the challenges posed by graft rejection and dysfunction. Ultimately, the strategic application of biomarkers has the potential to yield better long-term results for kidney transplant recipients and to deepen our understanding of the immunological mechanisms underlying rejection.

## Figures and Tables

**Table 1 medicina-61-00262-t001:** Comparative table of alloantibody detection tests [36,44,45,46,48].

Test/Assay	Description	Key Features	Limitations
Complement-Dependent Cytotoxicity (CDC) Crossmatch	Traditional test detecting antibodies to HLA by assessing complement-fixing antibodies in recipient serum that target donor lymphocytes.	- Detects complement-fixing antibodies. - Requires viable donor lymphocytes. - Positive when antibodies bind donor antigen and activate complement cascade.	- Only detects complement-fixing antibodies. - Dependent on the availability of viable donor lymphocytes.
Flow Cytometry Crossmatch	Utilizes flow cytometry to detect antibodies against HLA using fluorochrome-conjugated antibodies and donor lymphocytes.	- Detects low-titer IgG donor-specific antibodies (DSAs). - More sensitive than CDC crossmatch. - Aids in the early detection of graft dysfunction and antibody-mediated rejection [44].	- May not differentiate between pathogenic and non-pathogenic antibodies [44].
Solid-Phase Binding Assay (SPA)	Measures DSAs using single antigen beads for immunologic risk stratification, offering semiquantitative results through median fluorescence intensity (MFI) [36,45].	- Detects HLA antibodies using antigen-coated microbeads. - Provides semiquantitative MFI values. - Can guide desensitization protocols with treatments like PP/IVIG and immunosuppression [36,45].	- Susceptible to false positive or false negative results due to serum factors [36,45].
Panel Reactive Antibody (PRA) Assay	Evaluates sensitization status of patients awaiting transplantation by measuring antibodies against HLA [46,48].	- Critical for determining transplant eligibility. - Historically based on CDC methods. - Modern vPRA uses solid-phase immunoassays (e.g., single antigen beads) for enhanced precision [46,48].	- Traditional methods had low sensitivity and lacked the ability to distinguish between cytotoxic and non-cytotoxic antibodies [46,48].

**Table 3 medicina-61-00262-t003:** Biomarkers of rejection, allograft dysfunction, Ischemia–Reperfusion injury based on newer techniques.

Biomarker Type	Key Biomarkers	Assessment Methods	Pathogenic Characteristics/Function	Clinical Relevance
**Genes Overexpressed in the Common Rejection Module**	BASP1, CD6, CXCL10, CXCL9, INPP5D, ISG20, LCK, NKG7, PSMB9, RUNX3, TAP1	Microarray analysis, RNA-seq, qPCR, computational gene scoring	Activates signaling pathways, cytotoxicity, inflammation, immune reactions, T cell recruitment	Acute rejection, Chronic allograft dysfunction
**Transcriptomic Biomarkers**	4-gene model (vimentin, NKCC2, E-cadherin, 18S rRNA) 11-gene panel	Microarray analysis, next-generation sequencing	Increased metabolic activity, cellular stress, tissue damage, impaired repair mechanisms	Ischemia-Reperfusion Injury, Acute rejection, Fibrosis
**Epigenetic Biomarkers**	Foxp3 DNA demethylation PD1 DNA methylation microRNAs (e.g., miR-21, miR-200b)	qPCR, ChIP, single-cell techniques, flow cytometry, methylation analysis	Regulatory pathways in immune/inflammatory responses, reflecting non-coding RNA activity and methylation changes	Better graft outcomes, acute/chronic rejection
**Proteomic Biomarkers**	S100A8, S100A9, IL-6, IL-8, MCP-1, Cystatin-C	High-throughput proteomics (LC-MS, iTRAQ, etc.)	Inflammation, immune response, and tissue injury	Acute and chronic rejection
**Metabolomic Biomarkers**	NAD, 1-MN, cholesterol sulfate, GABA, nicotinic acid, NADPH, proline, spermidine	LC-MS/MS, HPLC, GC-MS	Inflammation, oxidative stress, immune responses, impaired tissue remodeling	TCMR, chronic allograft nephropathy
**Cellular Biomarkers**	Alloreactive CD8+ T cells (TEMRA and EM), CD154+ T cells, Tfh/Tfr ratio	Flow cytometry	Inflammation, cytotoxicity	Rejection, chronic allograft dysfunction

**Table 4 medicina-61-00262-t004:** Biomarkers in kidney transplantation, categorized based on their clinical utility.

Category	Biomarker	Characteristics/Function	Clinical Relevance
Diagnostic Biomarkers	Donor-derived cell-free DNA (dd-cfDNA)	Fragments of DNA from apoptotic donor cells	Detects graft injury and acute rejection, useful for antibody-mediated rejection (AMR).
	Donor-specific antibodies (DSAs)	Anti-HLA class I and II antibodies	Diagnoses AMR and aids in tailoring immunosuppressive therapy.
	Anti-MICA antibodies	Activate T cells and NK cells	Indicates acute and chronic AMR.
	Anti-AT1R antibodies	Linked to inflammatory responses and vascular injury.	Associated with AMR and graft survival outcomes.
	C4d	Marker of complement activation.	Indicates ongoing AMR processes.
	Gene expression profiles (GEP)	Measure gene expression in blood	Detects early phases of acute rejection, reducing unnecessary biopsies.
	MicroRNAs (e.g., miR-21, miR-155)	Altered expression patterns	Correlate with acute rejection.
	Plasma endothelial microparticles	Indicators of endothelial dysfunction	Early diagnosis of acute rejection.
	Urinary biomarkers (e.g., NGAL, KIM-1)	Reflect tubular injury	Predict acute rejection.
Prognostic Biomarkers	Intracellular ATP (iATP)	Measures immune function in T cells	Predicts infection or rejection risk.
	Donor-specific IFN-γ lymphocytes	Reflect immunological memory	Correlate with rejection risk.
	Soluble CD30 (sCD30)	Reflects T cell activation	Predicts graft outcome and alloimmune response.
	Torque tenovirus (TTV)	Reflects host immunosuppression levels	Mixed results in predicting rejection or infection.
	Fibrinogen alpha/beta/gamma chains	Reflect ischemia–reperfusion injury.	Useful for detecting rejection and injury.
	Complement C5a	High urinary levels indicate delayed graft function	Predicts impaired graft outcomes.
Therapeutic Biomarkers	dd-cfDNA	Indicates allograft injury	Guides treatment responses to rejection therapies.
	DSAs	Activate complement pathways	Help tailor immunosuppressive therapy.
	Anti-MICA and anti-AT1R antibodies	Promote immune cell activation and inflammation	Associated with graft injury and survival.
	Anti-VEGF antibodies	Disrupt VEGF signaling	May lead to endothelial injury and renal allograft rejection.
	Gene expression profiles (GEP)	Monitor gene activity	Identifies risk of subclinical rejection.
	Soluble CD30	Indicates T cell activation	Elevated levels predict poor outcomes.
	Torque Tenovirus (TTV)	Viral load reflects immune status	May predict infection or rejection risk.
	Plasma endothelial microparticles	Indicate endothelial dysfunction	Early markers for AMR.
	Extracellular vesicles (EVs)	Reflects kidney function statu	Detect acute rejection and assess graft health.
Monitoring Biomarkers	DSAs	Level guide therapy adjustments	Helps in monitoring rejection risks.
	dd-cfDNA	Indicates response to anti-rejection therapy	Tracks treatment effectiveness.
	Anti-VEGF antibodies	Indicates graft dysfunction risk	May inform therapy adjustments.
	Complement activation markers	Assesses ongoing rejection	Provides insights into complement-related injury.
	Urinary exosomal biomarkers	Includes CD30 and TIM-3	Indicates rejection status.
Traditional Biomarkers	Serum creatinine, eGFR, proteinuria	Indicates kidney function	May detect damage only after significant injury occurs.
Emerging Biomarkers	Genetic (e.g., GEPs, specific genes)	Identifies immune-related genes	Indicate rejection risk.
	Transcriptomics (4-gene, 11-gene models)	Measure gene activity linked to rejection and fibrosis	Predicts acute rejection or interstitial fibrosis.
	MicroRNAs (e.g., miR-21, miR-150)	Regulatory molecules alter expression patterns	Reflect acute rejection and inflammation.
	Protein biomarkers	Cytokines, chemokines, inflammatory mediators	Detect graft dysfunction or rejection.
	Metabolomic biomarkers	Reflect cellular processes	Provide insights into rejection-associated metabolic changes.
	Cellular biomarkers	Specific immune cells (e.g., T cells)	Indicate rejection-related immune responses.
	Epigenetic biomarkers	DNA methylation patterns	Reflect long-term rejection or adaptation processes.

## Data Availability

No new data were created or analyzed in this study.

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
