# Peer review of "Non-Invasive Biomarkers for Early Diagnosis of Kidney Allograft Dysfunction: Current and Future Applications in the Era of Precision Medicine"

_medicina, 2025, doi:10.3390/medicina61020262_

Round 1

Reviewer 1 Report

Comments and Suggestions for Authors

At the beginning, it is commendable to emphasize that there is no excessive overlap of the text beyond the 20% authenticity.

- In the abstract, list at least some innovative biomarkers or some omics technique!

- In the introduction, indicate that the outcome and length of allograft survival are also influenced by the length of dialysis, residual renal function, and cold ischemia time, especially in the part about allograft rejection!

- Biopsy of allograft is the gold standard, but it is necessary to emphasize other parameters in the introduction, especially when talking about TMA (ADAMST 13, LDH, haptoglobin, Coombs test).

- Do frequent thrombosis of vascular accesses speak in favor of a possible tendency to thrombosis?

Why is the importance of testing factors of tendency to thrombosis not emphasized?

- 2.2.3 In highly sensitized individuals, when the crossmatch is positive, is the use of PF and prednisone recommended and then repeated DSA check before Ts, especially in related living Tx?

- In Table 1, not a single autophagy factor is emphasized? Have any sets been investigated?

- When talking about fibrinogen chains - do you mean glycosylation or some other post-translational modification and which one?

- The omics techniques in Table 2 may be more appropriate to present schematically to make it clearer and easier to understand!

Author Response

Dear Reviewer,

Thank you sincerely for taking the time to review my article and for providing such valuable and insightful feedback. Your comments have been constructive in enhancing my understanding and clarity on kidney transplantation.

Below is the revised version of the manuscript from the original article, modified following your instructions:

-In the abstract, list at least some innovative biomarkers or some omics technique!

Thank you so much for this valuable comment, we have modified the abstract and listed novel biomarkers as well as the omics techniques, to identify them.

We have revised the abstract in response to your comment. Specifically, in line 30, page 1 of 24, of the peer-reviewed manuscript, the original text was removed and replaced with a new one, which has been highlighted in red for clarity.

-In the introduction, indicate that the outcome and length of allograft survival are also influenced by the length of dialysis, residual renal function, and cold ischemia time, especially in the part about allograft rejection!

Thank you so much for your thoughtful comment. We have revised the introduction to include factors that influence graft survival.

Below line 52, an addition has been made, highlighted in red, to address your important note. “The length of time on hemodialysis before……………………….. persistent inflammation, metabolic acidosis[11].”

-Biopsy of allograft is the gold standard, but it is necessary to emphasize other parameters in the introduction, especially when talking about TMA (ADAMS TS 13, Ldh, haptoglobin, coombs test)

Your comment has been instrumental in providing a more comprehensive perspective on the subject of diagnosing potential kidney allograft rejection.

In the introduction a paragraph has been added to address your very important note, which is highlighted in red. “Furthermore, thrombotic microangiopathy (TMA) is a critical pathological entity……………………………… of suspected thrombotic microangiopathy (TMA)[20].

-Do frequent thrombosis of vascular accesses speak in favor of a possible tendency to thrombosis?

Thank you for bringing attention to the omission of such a significant theme. We have revised the original text to address your valuable comment.

On page 3, a new paragraph has been added and highlighted in red to address your important note. Another crucial consideration to keep in mind is the fact that frequent thrombosis of vascular accesses…………………… elevated risk for thrombotic events, facilitating the implementation of targeted preventive measures[21]”

-In highly sensitized individuals, when the crossmatch is positive, is the use of PF and prednisone, recommended and then repeated DSA check before Ts, especially in related living tx?

The original text has been revised in response to your invaluable comment, specifically to address the issue of kidney transplantation in highly sensitized patients. In the section 2.2.3.1 Alloantibody detection tests include, there has been an addition, highlighted in red to address your important note.”Specifically, in cases of highly sensitized individuals who have a positive crossmatch…………………… This approach aims to reduce donor-specific anti-HLA antibodies (DSA) to allow for transplantation to occur[57].

-In Table 1, not a single autophagy factor is emphasized. Have any sets been investigated?

Once again, I sincerely appreciate your attention to a theme that had been overlooked. In response to your comment, we have revised the original text and incorporated the previously missing information. In table 2 (previous table 1), an additional line has been inserted below the existing table to meet the aforementioned requirement. Additionally, an explanatory paragraph has been added “Autophagy-related proteins commonly investigated……………………. provide insights into their potential involvement in graft injury or rejection[75]”

-When talking about fibrinogen chains – do you mean glycoslation or some other post-translational modification and which one?

Thank you for allowing the opportunity to clarify the aforementioned issue. The article cited in the review discusses fibrinogen beta (FGB) and fibrinogen gamma (FGG) in the context of their increased levels in urine associated with acute rejection (AR) of renal allografts. It does not specifically mention glycosylation or other post-translational modifications in relation to these fibrinogen chains.

-The omics techniques in Table 2 may be more appropriate to present schematically to make it clearer and easier to understand!

Thank you for highlighting the need for the article to be more comprehensive and accessible. We have made some revisions in accordance with your suggestion. We tried to describe all this information schematically, but we fount we loose data, therefore we revised the previous table accordant to your recommendations, and we believe it has been much easier to understand.

Reviewer 2 Report

Comments and Suggestions for Authors Thank you for inviting me to review the manuscript. I have thoroughly reviewed it with great interest.The title of the study effectively reflects the main objective, and the abstract is concise, summarizing the study conducted. Below are my comments:   1.  The content in lines 37–41 of the introduction section requires appropriate references to support the claims. Similarly, lines 48–52 also lack proper citations. Lines 60–66 are missing adequate references as well . Additionally, lines 133–143 are not supported by any references. The entire paragraph from lines 144–149 is supported by only one reference.Overall, the manuscript needs a thorough review to ensure proper referencing throughout.     2. The terms HLA-incompatible kidney transplantation and DSA-incompatible transplantation are closely related but refer to distinct aspects of immune incompatibility in kidney transplantation.  HLA-incompatible kidney transplantation : occurs when the donor's HLA are not fully matched to the recipient, and the recipient has pre-existing anti-HLA antibodies against the donor's HLA antigens. DSA-incompatible transplantation:This refers specifically to transplantation where the recipient has DSAs against the donor's antigens, whether these are HLA antigens or non-HLA antigens.   Authors can review and revise the content for improved clarity to help readers distinguish between these two terms effectively.   3. Please elaborate the information under the section 2.2.1.1 "HLA Epitopes Definition and Classification" for its role in immunological risk.   4. The authors should consider presenting the information on alloantibody detection tests in a comparative tabular format. Additionally, include details about the PRA assay in this section .   5. Though the Biomarkers have been extensively presented in the manuscript in greater detail. But  authors can "Group biomarkers" based on their clinical application (e.g., diagnostic, prognostic, therapeutic monitoring) for easier understanding.   Further, subheadings can be used to separate different categories, such as Donor-Derived Cell-Free DNA, Antibodies (HLA and Non-HLA), Urinary Biomarkers, and Proteomic and Genomic Biomarkers. Emphasize the sensitivity and specificity of biomarkers like dd-cfDNA, DSAs, and other non-HLA antibodies, as discussed by the reviewer. Provide a balanced view of the advantages and limitations of these biomarkers, as well as areas requiring further validation.   6.  Define technical terms like "i-IFTA," "ABMR," and "TCMR" on their first mention for readers who may not be familiar with them.

Author Response

Dear Reviewer,

We sincerely appreciate the time and effort you dedicated to reviewing our submitted article. The insights and recommendations you provided have been invaluable and have greatly contributed to a deeper understanding of the subject of kidney transplantation.

  • The content in lines 37-41 of the introduction section requires appropriate references to support the claims. Similarly, lines 48-52 also lack proper citations. Lines 60-66 are missing adequate references as well. Additionally, lines 133-143 are not supported by any references. The entire paragraph from lines 144-149 is supported by only one reference. Overall, the manuscript needs a thorough review to ensure proper referencing throughout.

We have thoroughly reviewed the article and revised the references following your recommendations. Thank you for emphasizing the significance of this aspect.

  • The terms HLA-incompatible kidney transplantation and DSA-incompatible transplantation are closely related but refer to distinct aspects of immune incompatibility in kidney transplantation and DSA incompatible transplantation are closely related but refer to distinc aspects of immune incompatibility in kidney transplantation. HLA-incompatible kidney transplantation: occurs when the donor’s HLA are not fully matched to the recipient, and the recipient has pre-existing anti-HLA antibodies against the donor’s HLA antigens. DSA incompatible transplantation: This refers specifically to transplantation where the recipients has DSAs against the donor’s antigens, whether these are HLA antigens or non-HLA antigens. Authors can review and revise the content for improved clarity to help readers distinguish between these two terms effectively.

We sincerely thank you for providing a deeper understanding of the subject of kidney allograft incompatibility through your comment. It has significantly contributed to gaining a clearer perspective on this aspect.

An explanatory paragraph has been added in the section 2.2.1 DSA-s incompatible transplantation “HLA-incompatible kidney transplantation…………….. DSAs are absent, indicating a distinct pathogenic role in transplantation[39].

  • Please elaborate the information under the section 2.2.1.1 HLA epitopes definition and classification for its role in immunological risk.

Thank you for kindly reminding me to include such a crucial aspect related to HLA molecules. A paragraph explaining the aforementioned topic at the end of section 2.2.1.1 HLA Epitopes Definition and Classification. “The classification and definition of HLA epitopes…………… to enhance transplant success and patient outcomes[43]

  • include details about the PRA assay in this section.

Thank you for your invaluable comment, which has contributed to a more comprehensive understanding of alloantibody detection tests. A paragraph has been added in the section 2.2.3.1. Alloantibody detection tests explaining this topic  is “The Panel Reactive Antibody (PRA) assay: An essential tool in transplant immunology ………………………..Combining desensitization with kidney-paired donation can be an effective strategy for transplanting sensitized patients and increasing transplant rates”

  • The authors should consider presenting the information on alloantibody detection tests in a comparative tabular format.

Thank you for enhancing the clarity of the article through your valuable advice. A new table (Table 1) that meets the above criteria has been added.

  • Emphasize the sensitivity and specificity of biomarkers like dd-cfDNA, DSAs and other non-HLA antibodies, as discussed by the reviewer.

Thank you for emphasizing the clinical significance of biomarker sensitivity and specificity. A paragraph addressing the above topic has been added under section 3.4 and highlighted in red for clarity. “3.4  Specificity and sensitivity of biomarkers. Presented below is a representative list of some of the most commonly…………………….. of anti-AT1R antibodies, which are associated with antibody-mediated rejection[99]

  • authors can group biomarkers based on their clinical application (e.g. diagnostic, prognostic, therapeutic monitoring) for easier understanding.

We sincerely thank you for your valuable contributions, which have significantly improved both the clarity and overall accessibility of this article, making it more comprehensible and engaging for its intended audience. A table (Table 4) that addresses the above recommendation has been added, under section 3.5, and highlighted in red, for clarity.

  • Provide a balanced view of the advantages and limitations of these biomarkers, as well as areas requiring further validation

A text that discusses the above criteria has been added on page 24 of 29, highlighted in red, under section 4, named as “Advantages and Limitations”.

We would like to express my heartfelt gratitude for the detailed and thoughtful feedback you provided through your review. The valuable insights and constructive suggestions you have offered have been instrumental in improving the clarity, coherence, and overall comprehensibility of our article

Reviewer 3 Report

Comments and Suggestions for Authors

Review for Medicina on the article:

“Non-Invasive biomarkers for early diagnosis of kidney allograft dysfunction. Current and future applications in the era of precision medicine”

Christina Lazarou 1, Eleni Moysidou 2, Michalis Christodoulou 2, Georgios Lioulios 3, Asimina Fylaktou 4 and Maria Stangou

The authors conducted a large revision of the literature concerning new biomarkers for early detection of kidney transplant dysfunction. The argument is relevant and of great interest for the whole nephrologist community. The chapters are punctuals and the subchapters cover all the aspects that need to be discussed.

The table and the description of the biomarkers is wide and easily readable.

I have only a few concerns:

- the authors didn't evidence that the type of dialysis pre-transplant could have an impact in the development of DSA or other Ab that could be harmful during transplantation.

- the authors adeguately describe all the biomarkers but they didn't provide a real opinion on what could be an important one for the future. The conclusions are not strong therefore I suggest to improve.

Author Response

Dear Reviewer,

we are sincerely grateful for your valuable contributions, which have greatly enhanced the clarity and depth of the analysis presented in our article. Your insightful comments were instrumental in fostering a deeper understanding of the subject discussed in the paper.

  • The authors didn’t evidence that the type of dialysis pre-transplant could have an impact in the development of DSA or other Ab that could be harmful during transplantation

A paragraph addressing the above recommendation has been added on page 2, which described excisting data regarding the impact of dialysis method in antibodies development. “Research has shown that patients undergoing pre-transplant PD………….. potentially impacting the development of DSA[12]”

  • The authors adequately describe all the biomarkers but they didn’t provide a real opinion on what could be an important one for the future. The conclusions are not strong therefore I suggest to improve.

Thank you very much for providing me with the opportunity to reflect on this significant question. We revised the conclusions in order to emphasize clinical importance in the future. We added a paragraph “Donor-specific antibodies (DSAs)………….. its levels can correlate with the severity of rejection[100]”

Round 2

Reviewer 1 Report

Comments and Suggestions for Authors

No

Reviewer 2 Report

Comments and Suggestions for Authors

Thank you to the authors for addressing the suggestions. I have no further comments.